Towards sustainable coastal management: aerial imagery and deep learning for high-resolution Sargassum mapping

Arellano-Verdejo Javier 1 javier.arellano@mail.ecosur.mx
http://orcid.org/0000-0002-5757-6081 Lazcano-Hernandez Hugo E. 2 hlazcanoh@ecosur.mx
1 Department of Observation and Study of the Earth, Atmosphere, and Ocean, El Colegio de la Frontera Sur , Chetumal, Quintana Roo , Mexico
2 Department of Observation and Study of the Earth, Atmosphere and Ocean, CONAHCYT-ECOSUR , Chetumal, Quintana Roo , Mexico
Wilson Matthew
Electronic publication date: 2024 Sep 23
Publication date: 2024
Volume: 12
Electronic Location ID: e18192
Received 2024 Mar 7; Accepted 2024 Sep 6
Copyright: © 2024 Arellano-Verdejo and Lazcano-Hernandez
Copyright year: 2024
Copyright holder: Arellano-Verdejo and Lazcano-Hernandez
License: This is an open access article distributed under the terms of the Creative Commons Attribution License, which permits unrestricted use, distribution, reproduction and adaptation in any medium and for any purpose provided that it is properly attributed. For attribution, the original author(s), title, publication source (PeerJ) and either DOI or URL of the article must be cited.
License URL: https://creativecommons.org/licenses/by/4.0/

Keywords: Beach monitoring, Artifitial neural network, Drone imagery, Remote sensing, Algal bloom, Geographical information system, Caribean sea

Funding: The authors received no funding for this work.

==============================
The massive arrival of pelagic Sargassum on the coasts of several countries of the Atlantic Ocean began in 2011 and to date continues to generate social and environmental challenges for the region. Therefore, knowing the distribution and quantity of Sargassum in the ocean, coasts, and beaches is necessary to understand the phenomenon and develop protocols for its management, use, and final disposal. In this context, the present study proposes a methodology to calculate the area Sargassum occupies on beaches in square meters, based on the semantic segmentation of aerial images using the pix2pix architecture. For training and testing the algorithm, a unique dataset was built from scratch, consisting of 15,268 aerial images segmented into three classes. The images correspond to beaches in the cities of Mahahual and Puerto Morelos, located in Quintana Roo, Mexico. To analyze the results the fβ-score metric was used. The results for the Sargassum class indicate that there is a balance between false positives and false negatives, with a slight bias towards false negatives, which means that the algorithm tends to underestimate the Sargassum pixels in the images. To know the confidence intervals within which the algorithm performs better, the results of the f0.5-score metric were resampled by bootstrapping considering all classes and considering only the Sargassum class. From the above, we found that the algorithm offers better performance when segmenting Sargassum images on the sand. From the results, maps showing the Sargassum coverage area along the beach were designed to complement the previous ones and provide insight into the field of study.

Introduction

Sargassum is a widespread macroalgae found in oceans worldwide, the records of 351 species have been documented (Guiry et al., 2014). Most species are benthic, i.e., they live on the seafloor. Only the species Sargassum natans and Sargassum fluitans, which from now on will be referred to as Sargassum, are holopelagic, i.e., they float freely on the sea surface throughout their life cycle (Dawes & Mathieson, 2008; Amaral-Zettler et al., 2017). Sargassum is usually distributed in the Sargasso Sea, the north Caribbean Sea, and the Gulf of Mexico, forming a clockwise “Sargassum migratory loop system” (Webster & Linton, 2013). In the open ocean, Sargassum rafts are a unique floating ecosystem in oceanic waters, which are generally poor in substrates and nutrients (Amaral-Zettler et al., 2017). As a result, floating Sargassum hosts a wide variety of invertebrate, fish, and turtle species (Coston-Clements et al., 1991; Rodríguez-Martínez, van Tussenbroek & Jordán-Dahlgren, 2016), providing ecosystem services such as refuge, feeding, and breeding areas for numerous species of high ecological and commercial interests (Rooker, Turner & Holt, 2006; Witherington, Hirama & Hardy, 2012).

On the other hand, Sargassum becomes a challenge when it accumulates in large quantities along the beaches, as it begins to unbalance the carrying capacity of ecosystems (Del Monte-Luna et al., 2004), occupying space and resources. For example, the large amount of biomass on the surface hinders the passage of light, affecting the flora (Van Tussenbroek et al., 2017) and fauna (Rodríguez-Martínez et al., 2019) of the seabed. When this large amount of organic matter enters a state of decomposition, it favors eutrophication and water pollution, affecting coastal ecosystems (Van Tussenbroek et al., 2017), the health of the inhabitants (Resiere et al., 2018; Fraga & Robledo, 2022), and the socioeconomic balance of the surrounding communities (Martínez-González, 2019; Chávez et al., 2020). For all these reasons, it is useful to be able to map and quantify Sargassum along the beach, one option to address this task is by proximal remote sensing.

Sargassum ocean observation from space began in 2005 analyzing imagery from the sensors MERIS-Envisat, and MODIS-Terra/Aqua through the indices Maximum Chlorophyll Index (MCI), and Fluorescence Line Height (FLH), traditionally used for Chlorophyll a monitoring (Gower et al., 2006). The analysis of the images provided by satellite platforms using novel indices began in an attempt to highlight floating objects in the ocean. For instance the Floating Algae Index (FAI) (Hu, 2009) and the Alternative Floating Algae Index (AFAI) (Wang & Hu, 2016) have helped monitor Sargassum at mesoscale and synoptic scales and have helped to observe its presence in the Atlantic Ocean (Wang & Hu, 2016), enabling the study of large Sargassum arrivals along the coasts of the Caribbean Sea in recent years (Wang et al., 2019). However, due to various atmospheric phenomena, as well as the technical characteristics of sensors and satellite platforms, it is not always possible to have the desired scenario for studying and monitoring Sargassum in the open ocean (Lazcano-Hernandez, Arellano-Verdejo & Rodríguez-Martínez, 2023). For instance, atmospheric humidity, clouds, intense solar reflections, water turbidity, and African dust are some factors that hinder the observation of Sargassum in satellite images (Wang & Hu, 2016). In addition, satellite platforms have a defined periodicity in their orbit and height above the Earth, so that the temporal and spatial resolution of the images are fixed for each satellite sensor. On the other hand, from the spectral point of view, the atmosphere hinders the observation of some wavelengths associated with chlorophyll fluorescence with peaks at 685 and 740 nm, reflected by vegetation (IR) (Lazcano-Hernández et al., 2019). For example, water vapor is a filter at 730 nm, and oxygen shows absorption at 687 and 760 nm (Abbott & Letelier, 1999). Because of the above, having a useful image of the area under investigation at the desired time is not always possible.

In addition to the atmospheric and technical challenges already discussed in ocean monitoring through the analysis of satellite images using ocean color indices, the observation of Sargassum on the coast from space is compounded by the biodiversity of the region and the transparency of shallow waters. This increases the radiation reflected or scattered by the areas surrounding the pixel of interest to also be captured by the sensor and associated with that pixel as its own. This phenomenon is known as adjacency (Wang & Hu, 2016), which generates a lower contrast of Sargassum with the surrounding environment, making its observation and monitoring more complex. This is why, despite their usefulness in the open sea, in coastal waters, ocean color indices such as FLH, FAI, and AFAI are less able to highlight the Sargassum from the surrounding background, causing a drastic reduction in their usefulness under such conditions (Lazcano-Hernandez, Arellano-Verdejo & Rodríguez-Martínez, 2023).

To address the challenges faced in the monitoring of Sargassum in nearshore waters, recent studies have applied deep learning (DL) techniques to provide solutions from a different perspective and reduce false positives when applying ocean color indices. For instance, in the year 2019, a convolutional neural network (CNN), named ERISNet, was proposed for MODIS pixel classification (Arellano-Verdejo, Lazcano-Hernandez & Cabanillas-Terán, 2019); in 2020, the trainable nonlinear reaction-diffusion (TNRD) denoising model was implemented to minimize the noise in the MSI-FAI images, to facilitate the extraction of Sargassum features within the imagery (Wang & Hu, 2020); in 2021, a CNN named VGGUnet was proposed to automatically detect and quantify Sargassum macroalgae from various high-resolution multi-sensors (Wang & Hu, 2021); and in the year 2022, a u-net (Ronneberger, Fischer & Brox, 2015) was adapted to extract Sargassum features from Dove imagery along beaches and nearshore waters (Zhang et al., 2022). However, due to the dynamics of Sargassum stranding and the aforementioned limitations of satellite remote sensing, these solutions are insufficient and should be complemented by aerial imagery at other scales and in-situ information to help validate satellite information and fill information gaps.

On the other hand, Sargassum monitoring on beaches increased in 2018 due to Sargassum blooms on several beaches in the Caribbean. To complement the data provided for satellite imagery, the scientific community incorporated several techniques for data collection along the beaches; for instance, citizen science (Álvarez-Carranza & Lazcano-Hernández, 2019; Arellano-Verdejo & Lazcano-Hernandez, 2020; Iporac et al., 2019; Putman et al., 2023) and network camera systems (Rutten et al., 2021). This information enabled the building of datasets to train DL algorithms capable of classifying photographs with/without Sargassum (Arellano-Verdejo & Lazcano-Hernández, 2021), classifying beach photographs with or without beach perspective (Santos-Romero et al., 2022), and segmenting Sargassum into snapshots (Arellano-Verdejo, Santos-Romero & Lazcano-Hernandez, 2022) to create Sargassum coverage maps from beach-level photographs. The semantic segmentation technique in Fig. 1 has been one of the most widely used DL methodologies for identifying Sargassum within the collected beach-level photographs (Valentini & Balouin, 2020; Balado et al., 2021; Arellano-Verdejo, Santos-Romero & Lazcano-Hernandez, 2022). Table 1 shows a summary of some characteristics of the most recent publications using computer vision techniques for the classification or semantic segmentation of macroalgae within photographs.

Figure 1 Semantic segmentation is a technique in computer vision that involves labeling each pixel of an image with a category or class, allowing a detailed understanding of its visual content.

In the image, you can see (A) the source image and (B) the segmented image with five classes: sand (yellow), Sargassum (brown); water (cyan), vegetation (green), and sky (light blue).

Table 1 Some features of recent publications using computer vision techniques for the classification or semantic segmentation of macroalgae in photographs taken along beaches.

Paper	Task	CNN architecture/Dataset/Final product	
Valentini & Balouin (2020)	Coastal image classification	MobileNetv2/Dataset available from the authors/Image classifier	
Balado et al. (2021)	Semantic segmentation of images of five different macroalgae	Mobilnetv2, Resnet 18, and Xception/Dataset available from the authors/Semantic image segmenter	
Arellano-Verdejo & Lazcano-Hernández (2021)	Classifying images with/without Sargassum	AlexNet/https://doi.org/10.6084/m9.figshare.13256174.v5 Sargassum presence/absence map	
Santos-Romero et al. (2022)	Classifying imagery with or without beach perspective	MobileNetv2/Dataset available from the authors/Image cassifier	
Arellano-Verdejo, Santos-Romero & Lazcano-Hernandez (2022)	Segmenting Sargassum into snapshots	Pix2pix/https://doi.org/10.6084/m9.figshare.16550166.v1 Percentage of Sargassum coverage map	

Although beach photographs are not affected by all the atmospheric phenomena that affect satellite images, the environmental conditions of the photographic capture, such as natural lighting, shadows, and the presence of additional objects in the scene, can influence the quality of the image. Additionally, in citizen science, the sensors in cameras and smartphones used by volunteers and researchers could have different technical features. Therefore, the collected images will vary in aspects such as sharpness, white balance, blur, or noise. These differences are a challenge for the algorithms that must process and analyze the photographs. However, artificial neural network algorithms and computer vision offer novel solutions to address these challenges. It is possible to train models capable of performing image classification efficiently, even when image quality varies considerably. These models can be trained using sets of labeled images, with examples of the presence of Sargassum and clean beach images, captured under several conditions and with different devices, allowing the algorithms to learn to identify patterns and distinctive features of the Sargassum (Arellano-Verdejo & Lazcano-Hernández, 2021). In addition to the type of architecture, the performance of an algorithm depends on the quality of the dataset, in the context of Sargassum monitoring, the study (Arellano-Verdejo & Lazcano-Hernandez, In press) analyzes the impact of image quality on five neural network architectures; to evaluate the quality of the images the study uses the metrics of blur, BRISQUE and entropy.

In-situ monitoring of Sargassum is important to validate remote sensing results and to fill information gaps, but it also faces several limitations that are sometimes unavoidable. These limitations include the presence of Sargassum along hundreds of kilometers of beaches, the limited availability of electrical and Internet infrastructure needed to install surveillance camera networks, the need for a budget to cover the costs of maintenance and monitoring to maximize the useful life of the equipment and prevent looting; the legislative work for the design of surveillance systems on beaches, as the construction of infrastructure in protected natural areas, may be prohibited or must go through a rigorous approval process. On the other hand, citizen science data collection has low participation rates, and there is also a need to consolidate platforms for information exchange at the beach level, considering standardized metadata. To meet the challenges mentioned above, it is advisable to incorporate methodologies that help make measurements, at a local scale, in a faster and more efficient way. In this sense, the use of an unmanned aerial vehicle (UAV), commonly known as a drone, is considered a viable alternative to complement in-situ and satellite remote sensing observations (Chen et al., 2023).

The contributions of this study to the state-of-the-art include the construction of the first dataset of very high-resolution segmented aerial imagery for beaches with accumulated Sargassum. This dataset provides highly representative of coastal conditions, and it can be employed for the evaluation and comparison of different semantic segmentation algorithms in the context of Sargassum, serving as a benchmark for measuring the performance and effectiveness of the approaches. The semantic segmentation of Sargassum in aerial orthophotos allows the area calculation of Sargassum coverage along the beach in square meters and the design of high-resolution maps of Sargassum coverage.

Theoretical framework

Semantic segmentation

Semantic segmentation is an advanced technique in the field of computer vision that plays a key role in a variety of applications, from autonomous driving (Rizzoli, Barbato & Zanuttigh, 2022) to object detection in medical images (Asgari Taghanaki et al., 2021). As shown in Fig. 1, semantic segmentation involves assigning a semantic label to each pixel in an image, allowing objects or regions of interest within the image to be identified and differentiated. In the context of Sargassum detection, we can accurately distinguish Sargassum from other features in a coastal image, such as sand and water (Arellano-Verdejo, Santos-Romero & Lazcano-Hernandez, 2022).

One of the most widely used architectures in semantic segmentation, based on convolutional neural networks (CNNs) is the u-net (Ronneberger, Fischer & Brox, 2015). U-net is a type of CNN proposed by Ronneberger, Fischer & Brox (2015). The u-net’s name is derived from its shape reminiscent of the letter “U” as observed in Fig. 2. The unique u-net architecture has become a benchmark in the DL community for semantic segmentation tasks and has been adopted and extended in a variety of applications, from medicine (Huang et al., 2020) to robotics (Teso-Fz-Betoño et al., 2020).

Figure 2 A u-net is a convolutional neural network architecture designed specifically for semantic segmentation tasks in images, characterized by its U-shaped structure.

The u-net consists of two main parts: an encoding phase that gradually reduces the spatial resolution of the image and a decoding phase that restores it to its original resolution allowing the network to capture both high-level features and fine details. The u-net uses skip connections that connect layers from the encoding phase to the corresponding layers in the decoding phase. These connections allow information to flow directly from the encoding layers to the decoding layers, which assists in preserving important details during segmentation. The main contribution of u-net lies in its ability to effectively handle semantic segmentation, even in scenarios with low training data availability. Its skip connections, symmetric architecture, and customizable loss functions make it adaptable to diverse tasks.

On the other hand, pix2pix is a conditional generative adversarial network (cGAN) based on a u-net (see Fig. 3). Pix2pix was developed by researchers at the University of California, Berkeley, in 2016, (Isola et al., 2017). The cGAN is a variant of GANs (Generative Adversarial Networks), in which image generation from a given input is conditional. In the case of pix2Pix, this input is an image in an original domain that will be transformed into an image in a target domain.

Figure 3 Pix2pix is a conditional generative adversarial network (cGAN) used to transform images from one original domain to another, allowing the translation and generation of realistic and detailed images.

The pix2pix architecture consists of two main parts: the generator and the discriminator. The generator takes the input image in the original domain and produces an image in the target domain. The discriminator acts as a critic that evaluates whether an image is real (produced by the generator) or true (from the training dataset). During training, the generator and the discriminator engage in a zero-sum game, in which the generator tries to generate images that fool the discriminator, and the discriminator tries to identify the generated images.

Pix2pix has proven its versatility in various applications, such as image restoration (Pan et al., 2020), image translation (Zhu et al., 2017), artwork creation (Xue, 2021), and semantic segmentation (Arellano-Verdejo, Santos-Romero & Lazcano-Hernandez, 2022). One of the main advantages of pix2pix is its ability to work with unbalanced and relatively small datasets with a few thousand images (Isola et al., 2017). In the present study, pix2pix has been employed to perform semantic segmentation of images with the presence of Sargassum.

Materials and Methods

Pix2pix settings

For the pix2pix algorithm used in this study, the encoder implemented convolutional modules (C) with the subsequent distribution: C64 - C128 - C256 - C512 - C512 - C512 - C512 - C512 - C512, employing LeakyReLU activation functionality. The numeral ensuing the letter C specifies the number of filters employed in the convolutional layer, as indicated in Isola et al. (2017). The initial block (C64) utilized a batch normalization function. The decoder, conversely, utilized convolutional modules with the subsequent distribution: CD512 - CD512 - CD512 - CD512 - C512 - C256 - C128 - C64 were used with a ReLu activation function. The first three CD512 blocks incorporated a Dropout function with a probability of 0.5. All networks were trained from scratch. Layer weights were initialized from a Gaussian distribution with a mean of 0 and a standard deviation of 0.02. However, the discriminator network implemented a convolutional (C) architecture with the following distribution: C64 - C128 - C256 - C512 with a LeakyReLU activation function. As with the generator, the layer weights were initialized using a Gaussian distribution with a mean of 0 and a standard deviation of 0.02. The generator loss function was then estimated by calculating the average absolute error between the u-net output and the target image. The discriminator’s loss value was determined by summing the discrepancies between the actual value and the value detected by the discriminator, along with the disparities between the discriminator’s detected false values. The network underwent 30 generations of training.

Table 2 shows the number of parameters that are part of the model used to perform the semantic segmentation. As demonstrated in the table, the number of parameters of the generator is higher than the total number of parameters of the discriminator, which is not surprising given the pix2pix architecture. It is worth mentioning that not all parameters needed to be adjusted during the training phase.

Table 2 Number of pix2pix model parameters.

The pix2pix model is composed of two components: the generator and the discriminator. The parameters of each component are adjusted during the training phase.

	Pix2pix	
	Generator	Discriminator	
Trainable parameters	54,415,043	2,768,705	
Non-trainable parameters	10,880	1,792	
Total		57,196,420	

The pix2pix algorithm was implemented using the Python 3.10.9 programming language and the TensorFlow 2.10 library. All tests were performed using the Windows 10 operating system running on an Intel i5-6300HQ CPU @ 2.30 GHz, with 32 GB of RAM. An Nvidia GeForce GTX 950 M card was used to train the algorithms and the training time was approximately 12 h.

Dataset

The training dataset is critical in DL algorithms since it provides the basis for a model to acquire knowledge and learn to perform specific tasks. This data allows the model to identify patterns, relationships, and features in the provided examples, enabling it to generalize and make accurate decisions on new and unseen data during the training phase. The quality and representativeness of the training dataset have a direct impact on the performance and generalization capability of the model, so careful selection and preparation of this dataset is critical in the successful training of DL algorithms.

At the time this study was conducted, there were several open-access datasets used to train many of the state-of-the-art semantic segmentation algorithms. One of the most widely used is the COCO (https://cocodataset.org) (Common Objects in Context) dataset, which offers a wide range of images that include detailed semantic segmentations of objects in various contexts. COCO serves for both object detection and semantic segmentation. KITTI (https://www.cvlibs.net/datasets/kitti/) is another widely used dataset in perception and computer vision, specifically in applications of autonomous vehicles and driver assistance systems. This dataset comes from the KITTI project (Karlsruhe Institute of Technology and Toyota Technological Institute at Chicago) consisting of various image sequences and sensor data collected by a vehicle equipped with multiple cameras, lidar, and other sensors while driving in different urban and road environments. Finally, another dataset is Cityscapes (https://www.cityscapes-dataset.com/) which focuses on the semantic segmentation of urban scenes. It contains images of cities captured by vehicles equipped with cameras, and the labels include detailed segmentations of urban objects, such as streets, cars, and traffic signs. These are just a few examples of public datasets widely used in the field of semantic segmentation, and each one has its characteristics and challenges that make it valuable for training and evaluating algorithms in this area; however, in emerging problems, the lack of training datasets is one of the main issues.

The lack of datasets to train machine learning algorithms on emerging problems, such as Sargassum monitoring, is due to multiple reasons. First, these problems are often recent or unique, meaning that data collection may be limited or nonexistent. In addition, these issues can vary considerably by geographic location and time, making it difficult to create a generalized dataset. Data collection for addressing these problems can also be costly and logistically complicated; for example, the use of sensors in the ocean or conducting field surveys. In addition, the changing nature of emerging problems means that data initially collected may become obsolete as the situation evolves. In some cases, the need to label or annotate data can be a costly process. Finally, data are often held by government or private entities, making it difficult for public access when creating datasets.

The dataset used during the present study was elaborated from very high-resolution aerial images taken on the beaches of Puerto Morelos (SAMMO, 2020) and Mahahual, both in the state of Quintana Roo, Mexico. These images were taken on different dates and times of the day when diverse accumulations of Sargassum were present on the beach, as well as a variation in the lighting conditions, shadows, and the effects of the weather.

For the beach town of Mahahual, photographs were taken with a “DJI Air 2” drone between 10:00 and 14:00 h on April 21, 2021. Dronelink application supported autonomous flight (https://www.dronelink.com/). The main parameters flying were the following: flight altitude of 56.2 m, vertical displacement speed of 16 km/h, front overlap of 80%, side overlap of 70%, and picture shot rate of 2 s; from these parameters, the obtained GSD was m per pixel. For the beach town of Puerto Morelos, the images were provided by SAMMO (2020), and flights parameters can be requested through their customer service website (https://sammo.icmyl.unam.mx/).

To construct the dataset, five very high-resolution aerial images were used. As shown in Fig. 4 the images were processed using a windowing method with a 50% overlap. A reasonable overlap can reduce variability in model predictions by considering multiple perspectives of the input data. A smaller overlap could result in losing significant detail between windows, which could negatively affect model performance. A 50% overlap ensures that approximately half of the data in one window overlaps with the adjacent window. This allows for the model to see redundant information in both windows, which can assist in capturing relevant patterns and features within the data.

Figure 4 The very high-resolution image is divided into 256 × 256 pixel windows with a 50% overlap to assist in making the most of the input data information and contribute to robust and accurate model performance.

In some cases, this value can, however, be adjusted according to the specific needs of the problem or data type.

The final dataset consists of 15,268 RGB (red, green, blue) images, at 256 × 256 pixels each one. Using an 80/20 ratio, the images were divided randomly into two parts, namely the training dataset with 12,214 items, and the test dataset with 3,054 items.

The entropy (Eq. (1)) of an image is interpreted as a measure of the randomness or uncertainty present in the distribution of gray levels (or colors) in that image and is used as a metric for evaluating the information and content it provides. If the entropy is low, it means the image has a very predictable and uniform pixel distribution. In other words, there is significant repetition of gray levels or colors. This could indicate an image with homogeneous background areas. On the other hand, if the entropy is high, the image has a less predictable pixel distribution and more variability in the gray levels or colors, which could suggest the presence of details, edges, or diverse content in the image. Figure 5 shows the histogram showing the entropy of the dataset created for training the model. As demonstrated in Fig. 5, there are many images containing a low entropy, probably due to the fact that these images belong mainly to homogeneous areas such as sand, sea, and Sargassum, resulting in a low stress on the algorithm during the training phase, and causing a possible lack of generalization.

(1) H(X)=∑ip(xi)log2(1p(xi))

Figure 5 Entropy in an image measures randomness.

Low entropy implies a uniform pixel distribution, often in homogeneous backgrounds. High entropy suggests varied content. The dataset histogram reveals many low-entropy images, mainly in homogeneous areas like sand, sea, and Sargassum, which may limit algorithm generalization.

Due to the low variability of the aerial images, where the main element observed is water, the average value of the final entropy (indicated in red) is relatively low, which implies an unbalanced dataset from the information point of view, making the model training process and the segmentation task more complex.

Finally, in blue, we can observe the cumulative probability density of the entropy calculated for the final dataset. The cumulative probability density allows us to understand the probability with which the images are selected within the dataset at the time of training. As demonstrated in Fig. 5, there is a high probability of selecting images with a low entropy, which can cause the model to learn to identify patterns within the images with a uniform distribution of pixels. On the other hand, there is also a high probability of selecting images with a higher average entropy, which provides the algorithm with information about details that may be important when performing the final segmentation.

To show the number of pixels per class within the dataset, in Fig. 6, three histograms are presented. As it may be observed, the dataset contains many pixels with little information related to the classes Sargassum and sand. In the histograms on the left and in the middle, it is evident that few images clearly distinguish pixels with Sargassum and sand, which can be a challenge when learning to classify pixels of these classes. This is possible because the aerial images of the beach, by the nature of the scene, contain mostly water-related information. Finally, the histogram on the right shows that the pixels labeled as “other” do not present the same bias as the other two classes, which is related, again to the nature of the scene. This allows us to conclude that, due to the nature of the phenomenon and the images, the dataset will be consequently unbalanced, which again represents a challenge for classical segmentation algorithms.

Figure 6 Histograms reveal that most images have few pixels for “Sargassum” and “sand,” making them difficult to distinguish.

The nature of the phenomenon makes the dataset unbalanced, a challenge for segmentation algorithms.

Fβ-score metric

The f-score is a metric used in statistics and ML to evaluate the accuracy of a classification model. It is useful when dealing with binary classification problems, where one tries to predict whether an item belongs to one of two classes, for example, positive or negative. The f-score combines two metrics: precision and recall. These metrics are used together because they compensate for each other and provide a more complete view of model performance.

Precision (Eq. (2)) measures the proportion of positive predictions made by the model that are correct. It is calculated as the number of true positives (TP) divided by the sum of true positives and false positives (FP). On the other hand recall (Eq. (3)) measures the proportion of TP cases that the model correctly identified and is calculated as the number of TP divided by the sum of TP and false negatives (FN).

(2) precision=TPTP+FP

(3) recall=TPTP+FN

The f-score is relevant because it provides a balanced measure of the performance of a classification model, considering both the model’s ability to make correct predictions (accuracy) and its ability to identify all relevant cases (completeness). Depending on the application, the threshold for model decision-making can be adjusted to balance accuracy and completeness according to the needs of the problem. For example, in medical problems, it is crucial to maximize completeness so as not to lose positive cases, even if that means accepting some false positives. In other cases, such as spam detection, it is more important to maximize accuracy, even if that may result in lower completeness. The f-score assists in finding the right balance.

The f-score value varies between 0 and 1, where a higher f-score indicates better model performance. An f-score of 1 means that the model has perfect precision and recall, which is rare in practice. The f-score is a particular case of f-β score when β takes the value of 1. The f−β score is computed as shown in Eq. (4).

(4) f-β=(1+β2)⋅precision⋅recallβ2⋅precision+recall

where:

β is a positive value that determines the weighting of precision relative to recall. A value of β=1 results in an f1-score, while β>1 gives greater importance to precision, and β<1 gives higher importance to recall. Note that when the β value is equal to zero, the metric becomes the precision.

The f-β score is used when more control is needed over how accuracy is weighted compared to recall. For example, in problems where the priority is to minimize FPs, even if that means lower recall, a β value less than 1 could be used. On the other hand, in problems where you want to minimize FNs even if that results in more FPs, a β value greater than 1 could be used. To evaluate the effectiveness of the model presented in this study two values were used for β: β=0.5 and β=2 (Eqs. (5) and (6)).

(5) f0.5=1.25⋅precision⋅recall0.25⋅precision+recall

(6) f2=5⋅precision⋅recall4⋅precision+recall

The f0.5 score emphasizes the precision metric, which is valuable when we want to minimize false positives by ensuring that Sargassum detections are truly accurate and not confused with other elements such as sand or water. This is essential to avoid false positives that could lead to misinterpretation of the results. On the other hand, the f2 score gives more weight to recall, which is crucial in Sargassum detection, as we want to minimize false negatives and ensure that as much Sargassum as possible is correctly identified. This is important in applications where complete detection of Sargassum significantly involves the management and mitigation of its impact.

Figure 7 shows the value for f0.5 and f2 metrics as a function of the FPs and FNs. Assuming that the value of TNs does not change, in Fig. 7A, the value of the f0.5 and f2 metrics decreases as FPs and FNs increase. In terms of our study, the above would mean that the model is failing to classify the pixels. Therefore, pixels in the “Sargassum” class would be classified as the “other” class, while pixels in the “other” class would be classified in the “Sargassum” class.

Figure 7 (A–D) Metrics comparison.

Figure 7B shows the opposite case. If the value of the TN remains fixed, as the values of FP and FN decrease, the value of the metrics f0.5 and f2 increases, which means that the model is classifying the pixels correctly, and most likely, the ratio of FP and FN are similar. Thus the system has achieved a trade-off by minimizing FP and FN, a desirable scenario. Note that the curves for the cases shown in Figs. 7A and 7B are overlapped.

Figure 7C shows the case where the FP values grow, and the FN values remain unchanged. In this case, the value of the f2 metric is greater than the value of the f0.5 metric. This means that the system is classifying pixels in the “others” class as if they were pixels in the “Sargassum” class. This would mean that the system is overestimating the number of pixels in the “Sargassum” class, which is undesirable.

Finally, in Fig. 7D, the opposite case is depicted (FN values increase and the FPs remain unchanged). In this case, the value of the f0.5 metric is greater than the value of the f2 metric, which means that the system is classifying pixels in the “Sargassum” class as pixels in the “other” class, which implies that the system is underestimating.

Resampling

The bootstrapping resampling method is a statistical technique used to estimate the distribution of a statistic of interest and provides a robust, data-driven perspective for evaluating supervised learning algorithms. Using the resampling technique, one can obtain more accurate estimates of model performance and better understand model variability under different conditions.

Bootstrap resampling offers several advantages over cross-validation in the evaluation of neural network performance. These advantages include the ability to provide more robust and less biased estimates of model performance, better capture of uncertainty in performance estimation, the ability to construct accurate confidence intervals, and reduced susceptibility to variations in training and test datasets. These features make the bootstrap a valuable tool in the evaluation and analysis of neural network model performance, especially in contexts where data variability and estimation accuracy are critical.

When calculating performance metrics, such as accuracy, recall, or F β-score, in a test dataset, bootstrapping can provide confidence intervals for these metrics. This is useful to understand the variability in model performance and to have a more accurate estimate of model quality. In this study, resampling was used to calculate confidence intervals for the F0.5-score performance metric from 2,000 samples, which we believe can be valuable for understanding the generalization capability of pix2pix because, by calculating performance metrics for each sample, the total error can be decomposed in terms of bias and variance.

Results and discussion

One of the main contributions of this study is to create the first dataset of very high-resolution segmented aerial imagery for beaches with accumulated Sargassum. The semantic segmentation dataset of Sargassum imagery developed provides highly representative and diverse, encompassing a variety of coastal situations and conditions. The semantic segmentation labels have been created with accuracy and consistency, ensuring reliable labels and detailed information on the location and extent of Sargassum in each image. In addition, it can be employed for the evaluation and comparison of different semantic segmentation algorithms in the context of Sargassum, serving as a benchmark for measuring the performance and effectiveness of the approaches.

The analysis and discussion of the results is presented in two phases. On the one hand, the general performance of the pix2pix model will be analyzed; for instance, how the model is capable of segmenting an image into all its classes (Sargassum, sand, and others). Then, the analysis of the algorithm performance specifically for the Sargassum class will be shown, and the differences concerning the general case will be discussed. Finally, the potential areas where the algorithm could be applied in real cases will be presented.

General performance of the pix2pix model

In the context of the present study, and to make use of the metrics derived from the confusion matrix, we will refer to the Sargassum pixels as “positives” and the rest of the pixels as “negatives”.

By using the trained pix2pix model, semantic segmentation was performed on the 3,054 images of the test dataset.

Table 3 shows the results for the metrics used to evaluate the pix2pix model. Comparing precision and recall results, we can see a difference between them. Both the mean and median values are higher in the case of precision. When the number of FP (pixels of the sand and other classes classified as Sargassum class) tends to zero, the precision metric tends to one, while for the recall, when the number of FNs (pixels of the Sargassum class classified as “sand” and “other”) tends to zero, the recall metric tends to one. The above shows that the pix2pix model tends to minimize the number of FPs concerning FNs. In short, the proportion of Sargassum pixels that pix2pix misclassifies is smaller than the number of Sargassum pixels it is unable to detect. Thus pix2pix tends to slightly underestimate the amount of Sargassum it can observe, which in our case is not serious, since it is known at what times of the year Sargassum is upwelling on the beaches, what we are looking for is a quantitative estimation of the surface area of Sargassum cover on the beach.

Table 3 Stats.

	Mean	Sd	Median	Mad	Max	
Precision	0.8926	0.1320	0.9355	0.0830	0.9999	
Recall	0.8469	0.1783	0.8990	0.1217	0.9999	
f0.5	0.8637	0.1620	0.9163	0.1028	0.9999	
f2	0.8450	0.1801	0.8984	0.1213	0.9999	

Given the context of this study, it is preferable that the system underestimate rather than overestimate. There is evidence that Sargassum arrival in the Caribbean occurs with greater intensity in the spring-summer season (Wang & Hu, 2016; Wang et al., 2019). Therefore, the fact that the algorithm underestimates the presence of Sargassum ( f0.5>f2) is not critical for its adequate management and handling. It is de facto known that the macroalgae are present, and what is sought is a quantitative estimate of the area of Sargassum coverage that avoids falling into the sensationalism of the phenomenon.

A better metric would consider the two previous cases, a trade-off between precision and recall. For this, the f0.5 and f2 metrics (f β score with values for β of 0.5 and 2.0) were used. Table 3 shows that the mean and median values are slightly higher for the case of an f0.5 score, confirming again that the system tends to slightly underestimate the results.

To show the dispersion of the results obtained by the pix2pix model when segmenting aerial images, a number of steps were followed. First, the performance of the pix2pix model was tested by segmenting the three classes (Sargassum, sand, and other), and then the same procedure was performed only for the Sargassum class. In both cases, precision, recall, and f β-score metrics were calculated for β=0.5 and β=2 to measure the relationship between TP, FP, and FN. Finally, to obtain the confidence intervals of the segmentation algorithm, the results were resampled using the bootstrapping technique.

Figure 8A shows the box plot where the dispersion of the resulting data for the precision and recall metrics can be seen when pix2pix segmented the three classes. The precision results were less dispersed, i.e., the interquartile range is smaller compared to the recall results. On the other hand, we can also see that the median precision was higher than the median obtained by the recall, which shows a first indication that pix2pix minimized false positives concerning false negatives, which is probably associated with an underestimation of the Sargassum pixels.

Figure 8 Box plots of the resulting data for; (A) precision and recall metrics, and (B) fβ2 and fβ0.5.

Figure 8B shows the results of the f0.5-score and f2-score metrics when pix2pix segmented the three classes. In this case, the interquartile range of both metrics was more similar. However, the results of the f05-score metric show a distribution with a slight positive skewness, at the same time that 75% of the data of the f05-score metric had a value higher than 75% of the data of the f2-score metric. This seems to be another indicator that pix2pix underestimated the results by minimizing the number of FPs produced.

Figures 9A and 9B show the results of precision, recall, and F β-Score metrics for each class segmented by pix2pix. As seen in Fig. 9A, the class with the lowest dispersion in the results, as well as the highest value for the median, was the sand class. This suggests that pix2pix most effectively minimized the total false positives for this class, which is an indicator that the algorithm has a stable behavior when segmenting these pixels and is probably due, among other things, to the high contrast between the sand pixels compared to the pixels of the rest of the classes.

Figure 9 Precision and recall for classes Sargassum, Sand, and Others.

The next class with the best values was the “other” class, its interquartile range was higher than “sand” class and lower than “Sargassum” class. The median value was close to the value of the “sand” class. As observed in the histograms shown in the dataset section, one would expect that, given the number of pixels in this class was higher than the rest, pix2pix would have more information during its training process, and this would be reflected in a greater generalization capacity. However, since the pixels of the other class, are composed of a wider diversity of elements (i.e., water, sky, palm trees), it provided a higher variability in the values of the components that made up their color, causing the algorithm to have additional stress when classifying this class.

Finally, Fig. 9A depicts the precision values for the Sargassum class, showing the highest dispersion, given that the number of images with Sargassum is the one with the least number of pixels, causing the algorithm to lack sufficient information during the training stage. However, this is to be expected, given the nature of the phenomenon of Sargassum upwelling on the beaches that have been analyzed during this study.

Figure 9B shows the results for the recall metric. This graph shows the behavior of pix2pix for FNs. The class with the lowest dispersion of the metric is the “other” class, which implies that pix2pix minimizes more effectively the FNs, having a positive impact at the time of generalization. In the same figure, once again, we can see that the class with the highest interquartile range was the Sargassum one. However, the median value is above for the sand class. This suggests that pix2pix confounded the Sargassum pixels to a lesser extent, i.e., given the precision and recall values for this class, the algorithm tended to underestimate the amount of Sargassum detected which, as discussed above, is better than if the algorithm overestimated these values. Overall, we see that pix2pix performs acceptably when segmenting aerial imagery. Since there is no similar work in the state of the art, we believe that the results shown so far can provide a baseline for comparing future algorithms.

Once the algorithm was evaluated in its entirety (segmentation of all the classes), F β-Score metrics calculation were performed with β=0.5 and β=2, only for the results yielded by the algorithm for the Sargassum class (Table 4). To do this the classes “sand”, and “others” were unified to form a single class, which was also called “others”. As expected, the median value decreased, and the interquartile range increased because the number of pixels of the new class “others” is several orders of magnitude higher than the total number of pixels of the Sargassum class. As demonstrated in Fig. 10, the values for the F0.5 and F2 metrics are similar. However, the median value for the case of the F0.5 metric is slightly higher than the value for F2, which confirms that pix2pix has a slight tendency to underestimate the presence of Sargassum pixels in the aerial images.

Table 4 β scores for Sargassum.

	Mean	Median	Sd	
f05	0.6369	0.7763	0.3320	
f2	0.6252	0.7285	0.3325	

Figure 10 f β metrics.

Figure 11 depicts two histograms showing the probability distribution of the F0.5-Score metric when pix2pix was used to segment the aerial images contained in the test dataset. Figure 11A shows the distribution of the F0.5 score for the case where pix2pix segmented the three classes. The mean is 0.8636, using a 90% confidence interval. The performance of the algorithm can vary between 0.8590 and 0.8684, which we consider to be a good result, considering that there can be values higher than 0.8684, which is reliable since, as discussed above, the algorithm tends to underestimate the amount of pixels detected as Sargassum. On the other hand, Fig. 11B shows the histogram for the distribution of the F0.5 score metric when pix2pix was used to segment the image into Sargassum and the “other” classes. These findings suggest that the algorithm performance has apparently decreased since the mean is now 0.6371, with a 90% confidence interval between 0.6249 and 0.6483. However, the decrease in the average does not mean that the algorithm is performing poorly, in this case, it is because of, the large number of pixels for the “sand” and “other” classes skewing the results. We believe that these last measurements demonstrate two things. First, the algorithm behavior is not random. Second, it sets a benchmark for the assessment of future Sargassum segmentation algorithms using aerial photographs.

Figure 11 (A and B) Bootstrapping f0.5.

Based on the f0.5−score values obtained by resampling the test dataset, considering all classes (depicted in Fig. 11A), Fig. 12 shows five images with f0.5score values within and outside the computed confidence intervals. The information in each column corresponds to each image; the first row shows the source image, and the second row shows the image segmented by the algorithm (Sargassum is colored in red). The third row shows the 3×3 confusion matrix corresponding to the image in each column, the values in the confusion matrix represent the number of pixels per class in an image (other, sand and Sargassum), the ground-truth data are represented on the x-axis, and the predicted ones on the y-axis. Finally, at the bottom of the figure, we can see the f0.5 and f2-score values for each segmented image.

Figure 12 Sargassum f0.5 metric consider all classes.

First, we will comment on some features that are common to several of the images, and later we will comment on specific features of each image. In all the images in Fig. 12, the values of the main diagonal (TP) are one or two orders of magnitude higher than their corresponding FP and FN; which means that, for these images, the algorithm has classified well the high amount of pixels. On the other hand, focusing on the Sargassum class, the f0.5 values are higher than the f2 values, suggesting that the algorithm tends to underestimate the Sargassum class in general. Figure 12A shows a value of f0.5=0.7969, which is between the minimum value and quantile 1, and the f0.5 values in Figs. 12B–12E are between quartile 1 and quartile 2.

It is important to remember that the “others” class is the most abundant in the dataset, followed by the “sand” class and finally the “Sargassum” class. This imbalance in the dataset implies that the generalization capacity of the algorithm is better for the “others” class and lower for the “sand” and “Sargassum” classes, which in turn affects the quality of the image segmentation.

In the segmented images corresponding to Figs. 12A and 12D, FNs are observed in dry vegetation, which can be explained due to the similar spectral characteristics between dry leaves and dry Sargassum. On the other hand, FNs are also observed in areas of dry Sargassum that were mistaken for palm leaves, which is another manifestation of the spectral similarity between dry vegetation and dry Sargassum. In the segmented images of Figs. 12A–12D, FP is observed in small areas of sand surrounded by Sargassum that fail to be identified as “sand,” perhaps due to the inability of the algorithm to distinguish very small areas of one class within another. On the other hand, in the segmented images of Figs. 12A and 12C, FPs are observed on the sand and along the upper left edge, where there is no Sargassum, for this type of FPs, we have not identified a reason. However, they could be addressed by ANN explainability studies, which are beyond the scope of this study.

Figure 12A presents a value of f0.5=0.7969, located to the left of the confidence interval ( f0.5=0.8590). In the confusion matrix, the highest values were observed in the main diagonal (TP), meaning that most of the pixels were correctly segmented for all classes. “Others” is the most dominant class, since it presents the highest number of TP (26,198), followed by the “sand” class (14,783), and finally, the “Sargassum” class (10,283). This is because the highest number of pixels in the “source” image corresponds to vegetation, which belongs to the “others” class, followed by the “sand” class, and finally the lowest number of pixels is from the “Sargassum” class.

Figure 12B has a value of f0.5=0.8549, which is close to the confidence interval value on the left side ( f0.5=0.8590). In the confusion matrix, the “Sargassum” class is the one with the highest number of TP, which is desirable since the Sargassum class is the most abundant in the source image. In the segmented image, we can see that the algorithm is properly segmenting Sargassum in the three different stages of the macroalgae; the fresh one that presents a golden color, the one in decomposition that exhibits a brown color, and the dried one that shows a dark gray color.

Figure 12C has a value of f0.5=0.8608, which is very close to the mean value calculated by resampling ( f0.5=0.8636). In the main diagonal, the highest values of the confusion matrix are observed, which means that a high amount of pixels were correctly segmented. In the segmented image, the shades of the small dunes were confused with Sargassum, possibly due to a confusion between the dark tones of the shadows and the dry Sargassum, thus generating PF on the sand.

Figure 12D has a value of f0.5=0.8733, which is to the right of the confidence interval ( f0.5=0.8684) on the right side. In this case, the highest values of the matrix on the main diagonal can be observed again, which means a high amount of correctly segmented pixels.

Figure 12E has a value of f0.5=0.9222 and lies to the right of the confidence limit on the right side. In this case, the scene shows Sargassum floating in the sea, which is why the “sand” class is not present in this image. Thus in the confusion matrix, zeros are observed in the boxes corresponding to the “sand” class. It is noteworthy that the algorithm manages to distinguish a large part of the seabed vegetation and does not confuse it with Sargassum; however, in the segmented image, FP patches are observed in the lower right, which could be due to the algorithm confusing the image resulting from the combination of the seabed and the reflections on the water surface with Sargassum. Finally, FN are observed on the shores of the Sargassum areas, which can be explained as another manifestation of the spectral similarity between the Sargassum and the sum of the effects caused by the seabed and the surface.

It is relevant to consider that the images of the sea surface are dynamic, and the scene of the following moments in time may be different from the previous ones. This occurs because the sea surface changes constantly, mainly due to sea currents and wind, which is one of the reasons why we do not use the results of this study for the design of Sargassum coverage maps in nearshore waters.

Based on the f0.5-score values obtained by resampling the test dataset (depicted in Fig. 11B), Fig. 13 shows five images with f0.5 score values within and outside the computed confidence intervals. The information in each column corresponds to each photograph and has the same structure commented for Fig. 12. A segmentation analysis of Sargassum was performed. For this purpose, the classes “Others” and “sand” were unified into one, denominated “Others”. Therefore, the matrix is 2×2 because there are only two classes (Others and Sargassum). Therefore, it should be noted that the values of FP, FN, and TP in the 2×2 matrices in Fig. 13 represent the sum of the values of the “other” and “sand” classes. We will first comment on some characteristics common among the images and then on the specific features for each image. The f0.5 values in Figs. 13A–13D lie between quartile 1, and quartile 2, and the f0.5 value in Fig. 13E lies between quartile 2, and quartile 3.

Figure 13 Sargassum f0.5-score metric consider only Sargassum class.

In Figs. 13A–13D, the “Others” class is the most abundant, which can also be seen in the confusion matrix of these images, where the highest values are in the TN (lower right quadrant). This means that most of the pixels of the “others” class were well segmented. In Fig. 13E, the “Sargassum” class is the most abundant, in this case, the highest values are found in the quadrant corresponding to the TPs (upper left quadrant).

Most of the PFs and FNs present in the images in Fig. 13 may have their origin in the algorithm’s limitation in discriminating elements that share some spectral similarity. On land, the PF and FN occur mainly between dry vegetation and dry Sargassum. On the other hand, in the shallow aquatic areas near the beach, in addition to the constant change in the brightness of the scene due to the waves, the various elements that can be seen on the seabed are another factor that contributes to the appearance of PFs and FNs, as they contribute to the changes in the water surface. Additionally, there are scenes in which the color of the sea changes to brown tones because of the decomposition of the Sargassum, this change in the color of the water diminishes the contrast between the sea and the Sargassum, making segmentation difficult.

Figure 13A presents a value of f0.5=0.4125, located to the left of the confidence interval ( f0.5=0.6249). In the source image, the water is the most abundant element in the scene; the segmented image shows that the algorithm fails to identify all the Sargassum located in the upper left part of the image. Because it underestimates the Sargassum class, the confusion matrix has low values for the TP (upper left quadrant). The tendency of the algorithm to underestimate the Sargassum class can also be deduced from the fact that the value of the f0.5-score is higher than the f2-score.

Figure 13B presents a value of f0.5=0.5675, located to the left of the lefthand confidence interval ( f0.5=0.6249). The source image shows that the “Other” class is the most abundant. The segmented image shows water pixels segmented as “Sargassum” (FP) and “Sargassum” pixels not segmented as such (FN). In the confusion matrix, we can observe that the amount of FP is less than the amount of FN (FP < FN), which suggests that, in this image, the algorithm overestimates the presence of Sargassum, comparing the value of the β metrics ( f0.5<f2) is another path to conclude the algorithm overestimation in this image.

Figure 13C shows a value of f0.5=0.6433, located within the confidence intervals and to the right of the mean ( f0.5=0.6371). In the original image, the sea looks brown due to the concentration of leachates in the water. This condition tends to reduce the contrast between the Sargassum and the sea; thus, reducing the segmentation capacity of the algorithm. In this case, the Sargassum has been underestimated ( f0.5>f2).

Figure 13D presents a value of f0.5=0.7055, located to the right of the righthand confidence interval ( f0.5=0.6483). In the source image, it is seen that the “others” class is dominant. In the segmented image, some elements of a boat were segmented as Sargassum (FP), and in the upper left part of the image, we can see that Sargassum that is not segmented as such (FN). Comparing the metrics, we conclude that Sargassum is overall underestimated in this image ( f0.5>f2).

Figure 13E shows a value of f0.5=0.7939 and lies to the right of the right-hand confidence interval ( f0.5=0.6483). In the confusion matrix, the highest values are located in the main diagonal, which means that a large number of pixels have been correctly segmented. However, since f0.5<f2, we concluded that there is an overestimation of Sargassum in this image. From the analysis of Figs. 12 and 13, we can conclude that, generally, the methodology proposed in this study shows its best results in the segmentation of the Sargassum on beaches.

Sargassum mapping

Sargassum monitoring on the beach has a relatively recent history. The study of Arellano-Verdejo & Lazcano-Hernandez (2020) states that crowdsourcing is helpful for building a geotagged pictures dataset, to identify the presence or absence of Sargassum along the beach. Later, the study conducted by Arellano-Verdejo & Lazcano-Hernández (2021) implements the ideas proposed by the crowdsourcing study previously mentioned, and using several beach-level geotagged photographs that were collected, a presence/absence beach Sargassum map (Fig. 14A) was built. Afterward, the study conducted by Arellano-Verdejo, Santos-Romero & Lazcano-Hernandez (2022) goes forward and builds a dataset composed of one thousand segmented-geotagged Sargassum beach-level pictures to retrain a pix2pix algorithm. The percentage of Sargassum coverage was calculated from every segmented-geotagged image. Based on this information and through a tessellation of polygons of 30×30 square meters, it was possible to compute the average percentage of Sargassum coverage for each polygon, and finally design a map with percentages of Sargassum coverage for the study area (Fig. 14B).

Figure 14 Sargassum beach map evolution, study area located in Mahahual, Quintana Roo, México.

Maps corresponding to April 21, 2021. (A) Presence/absence Sargassum map built from images of the beach provided through crowdsourcing, (B) Sargassum coverage map, obtained through calculating the percentage of Sargassum coverage using geotagged photographs. (C) Sargassum area map obtained using an aerial orthophoto. (Source credit to Holger Weissenberger).

The maps above used different elements to represent the presence of Sargassum on the beach. The map in Fig. 14A used points to represent the locations where photographs with information on the presence/absence of Sargassum were taken. The map (Fig. 14B) used polygons to depict through a color ramp, the percentage of coverage for each area, calculated by averaging the percentage of Sargassum in each photograph within each square that conforms the tessellates, both maps show information that can be useful but do not allow us to identify the areas where Sargassum is on the beach, which is the initial information needed to make volume estimations. It is in this direction that the present study makes one of its main contributions, proposing a quantitative and scalable methodology for calculating the area of Sargassum coverage on the beach using aerial photographs. Figure 14C shows the Sargassum area map resulting from using this methodology. In this case, the polygon selected to visualize the macroalgae has the shape corresponding to that delineated by the Sargassum and is highlighted by a color ramp in a hot spot stile.

Figure 15 depicts an overview of the study area and some approaches to observing details of the proposed map. Figure 15A shows an orthophoto corresponding to the study area taken between 10:00 and 14:00 h on April 21, 2021. To build the orthophoto, the Open Drone Map (ODM) software (https://www.opendronemap.org/webodm/) was used to process 422 photographs taken with a “DJI Air 2” drone. The main parameters used for the mission flies were the following: flight altitude of 56.2 m, vertical displacement speed of 16 km/h, front overlap of 80%, side overlap of 70%, and picture shot rate of 2 s; from these parameters, the obtained Ground Sample Distance (GSD) was 2 cm per pixel.

Figure 15 Mahahual, Quintana Roo.

(A) Sargassum coverage-area map and orthophoto of the study area, corresponding to April 21, 2021., Approach of (B) B zone, Sargassum area: 123m2, (C) C zone, Sargassum area: 92m2, (D) D zone, Sargassum area: 144m2, (E) E zone, Sargassum area: 82m2. (Source credit to Holger Weissenberger).

The coastline in the town of Mahahual has shallow waters and is bordered by a discontinuous reef to the east, approximately 120–400 m offshore. There are also mangroves, rocky areas, and the lagoon bottom, which is covered with seagrass dominated by T. testudinum (Camacho-Cruz et al., 2022). All of these add to the challenge of identifying Sargassum in this environment. Figures 15B to 15E show approaches of the areas within boxes B to E, depicted in Fig. 15A. Figure 15B shows a diagonal zone with the presence of Sargassum that contrasts on the left with sand and on the right with a zone of shallow transparent water. In this case, the Sargassum coverage area was calculated at 123m2. Figure 15C shows Sargassum in the upper zone, which contrasts with shallow transparent water, and what looks like seagrass in the lower part of the image. In this case, the Sargassum coverage area was calculated at 92m2. Figure 15D shows a diagonal zone with the presence of Sargassum on the left side of the image and another area with a smaller quantity in the upper-right zone rocks are also observed in the middle of both Sargassum zones and the lower part of the image. In this case, the Sargassum coverage area was calculated at 144m2. Finally, Fig. 15E shows a diagonal zone with the presence of Sargassum on the left side of the image, which contrasts with shallow transparent water, and what looks like seagrass on the right side of the image. In this case, the Sargassum coverage area was calculated at 82m2.

Downtown Puerto Morelos has a wave-dominated beach with turquoise water and white sandy beaches (Escudero et al., 2021). Therefore, the diversity of elements near the beach is lower than in Mahahual, so the segmentation of Sargassum on the beach presents fewer challenges, when compared to Mahahual’s beaches. Figure 16 depicts an overview of the study area and some approaches to observing details of the proposed map. Figure 16A shows an orthophoto from Puerto Morelos seaside on July 13, 2020 (SAMMO, 2020). Figure 16B shows Sargassum along the beach with high contrast to the sand. The Sargassum coverage area calculated in this image was 27m2. Figures 16C–16E show Sargassum along the beach with high contrast to the sand on the left and with a zone of shallow transparent water mixed with Sargassum leachate. In the case of Fig. 16C, the Sargassum coverage area was calculated at 169m2, in Fig. 16D at 95m2; and in Fig. 16E at 78m2.

Figure 16 Puerto Morelos, Quintana Roo.

(A) Sargassum coverage-area map and orthophoto of the study area, corresponding to July 13, 2020, Approach of (B) B zone, Sargassum area: 27m2, (C) C zone, Sargassum area: 169m2, (D) D zone, Sargassum area: 95m2, (E) E zone, Sargassum area: 78m2. (Source credit to Holger Weissenberger).

Conclusions

The present study contributes to the state-of-the-art by producing high-resolution maps of Sargassum coverage, using aerial photographs and the pix2pix algorithm by image semantic segmentation. These maps offer the possibility of measuring in square meters the amount of Sargassum that covers the beach, which makes them a benchmark for the state of the art of Sargassum monitoring on the beach. They are also useful for informed decision-making regarding the management, use, and final disposal of Sargassum.

The first contribution of this study is the construction of a dataset of aerial images segmented into three classes (Sargassum, sand, and others) for the training of semantic segmentation algorithms. The photos were collected at different times and correspond to beaches in the cities of Mahahual and Puerto Morelos, located in the state of Quintana Roo, Mexico. The resulting dataset is unique in its type, consisting of 15,268 segmented aerial images. Compared to previous studies for the same region is the first of its kind.

To characterize the training dataset, we calculated the entropy for each image and the total number of pixels per class across all imagery. The values calculated for the entropy indicate sufficient photos in the dataset with valuable information for algorithm training. Regarding pixel count, the results suggest an unbalanced dataset in terms of pixels within the class of interest, which poses a challenge for the algorithm; ideally, there would be an equal number of pixels for all classes.

The analysis of the results utilized the f0.5 and f2 metrics. From a Sargassum class perspective, the findings depict a general balance between the false positives and false negatives, with a slight inclination towards the latter, which means that the algorithm tends to underestimate the Sargassum, which is not harmful to the purposes of our study. To better understand the algorithm’s performance, we applied the bootstrapping resampling technique to the f0.5 metric results in two cases: considering all classes and considering only the Sargassum class. This allowed us to establish confidence intervals for the algorithm’s generalization capacity in each case. Observations indicated that the algorithm performs well in segmenting Sargassum images on sand. Although the algorithm demonstrates adequate performance in certain instances when segmenting Sargassum over water, as measured by the f0.5 and f2 metrics, we caution against employing it for segmenting images of floating Sargassum.

The second contribution of this study lies in the design of quantitative maps of Sargassum cover along the beach. Using of geographic information systems, the segmented orthophoto was vectorized, which allows, for calculating the area in square meters that Sargassum covers on the beach. The differentiating element in these maps is the perimeter of the polygons used to depict the Sargassum; they have the shape of the macroalgae accumulated along the beach, allowing them to validate satellite remote sensing measurements.

Furthermore, given the segmentation algorithm’s ability to isolate sand on the beach, it is feasible to calculate the portion of the beach covered by Sargassum, and the percentage of the beach that is clear. This can have a favorable impact on designing effective beach cleaning logistics. Due to its achieved generalization capacity, estimated through the bootstrapping technique described above, the findings demonstrate the potential of this approach to segment aerial orthophotos of various areas of interest, without requiring consideration in the training dataset.

Findings from this study suggest that the performance of the pix2pix algorithm depends on the quality of the training dataset. Thus, to improve the performance of the algorithm, this would require retraining with a balanced dataset, in terms of the three classes used in the study (Sargassum, sand, and others), and including a higher amount of Sargassum images in various contexts like water, soil, and leachate, which is a challenge for Earth observation studies, such as this one.

Supplemental Information

Supplemental Information 1 Code for the Pix2Pix Neural Network.

https://cocodataset.org

https://www.cvlibs.net/datasets/kitti/

https://www.cityscapes-dataset.com/

https://www.dronelink.com/

https://sammo.icmyl.unam.mx/

https://www.opendronemap.org/webodm/

We want to thank Servicio Académico de Monitoreo Meteorológico y Oceanográfico, from Instituto de Ciencias del Mar y Limnología, Puerto Morelos Q. Roo México, from the Universidad Nacional Autónoma de México, for providing the orthomosaics of SITE 01, from the period 2020-07-08 and 2020-05-15. In addition, we would like to thank Holger Weissenberger for his support in producing (Figs. 14–16). We also thank Martin Santos-Romero for his technical assistance on hand segmentation of the images. Finally, we would like to thank Deon Victoria Heffington for her help in the revisions and editions of this manuscript.

Additional Information and Declarations

Competing Interests

Author Contributions

Data Availability

The authors declare that they have no competing interests.

Javier Arellano-Verdejo conceived and designed the experiments, performed the experiments, analyzed the data, prepared figures and/or tables, authored or reviewed drafts of the article, and approved the final draft.

Hugo E. Lazcano-Hernandez conceived and designed the experiments, performed the experiments, analyzed the data, prepared figures and/or tables, authored or reviewed drafts of the article, and approved the final draft.

The following information was supplied regarding data availability:

The data is available at figshare: Arellano-Verdejo, Javier (2024). Aerial Segmented Sargassum Dataset. figshare. Dataset. https://doi.org/10.6084/m9.figshare.25320148.v4

The code is available in the Supplemental File.

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
