# Peer review of "Towards sustainable coastal management: aerial imagery and deep learning for high-resolution Sargassum mapping"

_PeerJ, doi:10.7717/peerj.18192_

## Round 0.1 · original submission · Major Revisions

Both reviewers have found that your paper addresses an important topic and has considered relevant techniques to address scientific gaps. However, I agree with both reviewers that major revisions are needed before the manuscript can be considered for publication. Please address each point in the reviewers’ reports. In particular:

1. The manuscript is not clear in many critical sections, including a poorly written methods section. A clearer justification for the choice of algorithm should be included, with reference to the literature.
2. The literature review is reasonable but could be improved, such as through adding a table to present existing literature more clearly. The main contribution of this work needs to be highlighted clearly in the introduction.
3. Issues regarding the data processing and accuracy assessment need to be addressed, particularly with respect to potential biasing of the results, and regarding major deficiencies in the evaluation of the results (classification accuracy assessment).
4. Results need to be presented more concisely and be synthesised to avoid repetition.

Reviewer 1 ·

Basic reporting

All comments have been added in detail to the last section.

Experimental design

All comments have been added in detail to the last section.

Validity of the findings

All comments have been added in detail to the last section.

Additional comments

Review Report for "Towards sustainable coastal management: Aerial imagery and deep learning for high-resolution Sargassum mapping" in PeerJ

1. Within the scope of the study, classification and semantic segmentation processes were carried out using deep learning to detect Sargassum from aerial images.

2. In the Introduction section, what Sargassum is, its importance, its difficulties and basically semantic segmentation are mentioned by giving Figure-1. In this section, it is recommended to add a literature table consisting of columns such as dataset, model used, results, advantages/disadvantages, etc., in order to present the literature more clearly. Thus, the current topic and the literature can be compared better. In addition, it is recommended that the difference of the study from the literature and its main contributions to the literature be added more clearly at the end of the Introduction section.

3. In the Materials and Methods section, the semantic segmentation and dataset used in the study are basically mentioned. In the semantic segmentation section, first the u-net architecture, then the Pix2pix deep learning, a conditional generative adversarial network based on u-net, and the customizations made are explained. Although there are many different deep learning models that can be used for semantic segmentation when the literature is examined, it should be explained more clearly why the Pix2pix model is preferred. In the study of this model, the developed originality point, algorithm and hyperparameters used need to be explained in detail.

4. 15,268 aerial images with a 3-class structure consisting of "Sargassum, Sand and Other" classes were used as the dataset. The dataset used in the study is sufficient in quantity, size and quality. It has been stated that the dataset distribution is randomly divided into 80% training and 20% testing. The results obtained in classification and segmentation problems are very dependent on the dataset. For this reason, it is very important how the dataset distribution is made and how the training, validation and/or testing sections are determined. In order to analyze the results more accurately, it would be better to perform cross-validation instead of random distribution of the dataset. Based on this, it should be stated more clearly why cross-validation is not preferred and/or the dataset distribution ratio (80:20) and the basis on which the images are selected.

5. There are serious deficiencies in the evaluation metrics regarding classification and semantic segmentation in the results section. For example, metrics such as ROC curve, AUC score, precision-recall curve for both the triple class (Sargassum, Sand and Other) and binary class (Sargassum and Other) that are required to analyze the classification results correctly are not included.

As a result, although the study is important in terms of the dataset used and the problem addressed, it is recommended to pay attention to the parts explained in detail above.

·

Basic reporting

The manuscript addresses an important topic, and relevant techniques have been considered to address scientific gaps. However, the manuscript is not clear in many critical sections, as described in the detailed comments. For example, the method section is poorly written, and the results section contains a lot of methodological text, while surprisingly, the method section includes background/results-related text.

The literature review is reasonably adequate but could be improved. Some citations lack clarity regarding their relevance. For instance, the relevance of image noise-related work to this study is not clearly explained.

Experimental design

Basic questions remain unanswered from a data processing perspective. For example, the entropy results indicate imbalanced datasets, yet no data cleaning/preprocessing techniques were applied to address this issue. Despite the abundance of sand and water pixels compared to Sargassum pixels, no steps were taken to improve the dataset, potentially leading to biased results. Since AI-based solutions rely on data quality, biases in the data can propagate to the results, as observed in this study.

The method section lacks a clear description, with excessive background/context building. By the time a reader reaches the method section, they should encounter a well-laid plan and clear implementation steps, which are currently lacking.

Validity of the findings

In terms of content repetition, the authors dedicate significant space to discussing false positives and negatives, which are important but overly reiterated in the conclusions. Each results figure seems to conclude the same findings repeatedly.

Additional comments

Some of the detailed comments are presented below, and the authors can utilize them as guidance to improve the manuscript throughout.

Detailed comments:
- Line 31: The sentence lacks a clear conclusion.
- Lines 49: Consider adding a couple of sentences to highlight the need for remote sensing solutions for mapping.
- Line 52: Clarify why the atmospheric correction methods are not useful in this context.
- Lines 68-69: Provide details on the significance of the mentioned time and its link to challenges.
- Lines 70-77: Address whether the adjacency effect, part of atmospheric correction, could be utilized here and explain why pixel size may or may not be a problem with regard to the adjacency effect.
- Line 92: Clarify how satellite imagery is supported.
- Figure 1: Label the classes depicted.
- Lines 103-117: This paragraph lacks a clear conclusion and suffers from poor structure.

- Method section: Start with a plain English summary of the proposed method, detailing how it addresses the scientific gaps raised in the introduction.

- Lines 132-155: Divide this text into methodological choices and implementation sections, moving methodological choice texts to the introduction and implementation details to the method section.
- Line 164: Explain why the mean and SD values are set as such.
- Lines 183-197: Clarify the relevance of these lines to the reader.
- Lines 207-214: The text appears to read more like a discussion section. Details about the sensor used, spatial resolution, and bands are missing.
- Figure 4: Remove or make transparent the "50%" labels for clearer viewing.

- Lines 244-260: Clarify the implications of the model learning from images with different entropy levels and discuss preprocessing steps to address biases.
- Line 292: Explain the significance of choosing beta values of 0.5 and 2.0.
- Lines 293-310: Justify why results are presented/discussed in the method section.
- Line 334: Provide a clearer scientific rationale for the statement.

---

## Round 0.2 · accepted · Accept

Thank you for the extensive revisions and responses to comments made by the reviewers. I consider that you have address all reviewer comments (including reviewer 2, which I checked) and I am happy with the current version of the manuscript, which is now ready for publication.

Reviewer 1 ·

Basic reporting

All comments have been added in detail to the last section.

Experimental design

All comments have been added in detail to the last section.

Validity of the findings

All comments have been added in detail to the last section.

Additional comments

Thank you for the revision. The responses to the reviewer comments and the changes made to the paper are sufficient.